# Electrical Contact Resistance of Large-Area Graphene on Pre-Patterned Cu and Au Electrodes

**DOI:** 10.3390/nano12244444

**Published:** 2022-12-14

**Authors:** Tomáš Blecha, Zuzana Vlčková Živcová, Farjana J. Sonia, Martin Mergl, Oleksandr Volochanskyi, Michal Bodnár, Pavel Rous, Kenichiro Mizohata, Martin Kalbáč, Otakar Frank

**Affiliations:** 1Faculty of Electrical Engineering, University of West Bohemia, Univerzitní 2795/8, 301 00 Pilsen, Czech Republic; 2J. Heyrovsky Institute of Physical Chemistry, Academy of Sciences of the Czech Republic, Dolejškova 2155/3, 182 23 Prague, Czech Republic; 3Department of Physical Chemistry, Faculty of Chemical Engineering, University of Chemistry and Technology, Technicka 5, 166 28 Prague, Czech Republic; 4Tesla Blatná a.s., Palackého 644, 388 01 Blatná, Czech Republic; 5Department of Physics, Helsinki University, P.O. Box 43, 00014 Helsinki, Finland

**Keywords:** graphene, contact resistance, transfer length method, graphene–metal contact

## Abstract

Contact resistance between electrically connected parts of electronic elements can negatively affect their resulting properties and parameters. The contact resistance is influenced by the physicochemical properties of the connected elements and, in most cases, the lowest possible value is required. The issue of contact resistance is also addressed in connection with the increasingly frequently used carbon allotropes. This work aimed to determine the factors that influence contact resistance between graphene prepared by chemical vapour deposition and pre-patterned Cu and Au electrodes onto which graphene is subsequently transferred. It was found that electrode surface treatment methods affect the resistance between Cu and graphene, where contact resistance varied greatly, with an average of 1.25 ± 1.54 kΩ, whereas for the Au electrodes, the deposition techniques did not influence the resulting contact resistance, which decreased by almost two orders of magnitude compared with the Cu electrodes, to 0.03 ± 0.01 kΩ.

## 1. Introduction

Carbon allotropes, such as graphene or carbon nanotubes, are advantageously employed in many applications [1], such as in sensing technology as active layers of sensors for monitoring environmental parameters [2], in electrochemical storage devices [3], in nanophotonics [4], or for the fabrication of graphene field-effect transistors (GFETs) [5,6]. In GFETs, the channel is made of graphene, with which contact is made using metal electrodes that constitute the source and drain. The contact resistance (*R*_C_) between the metallic contact and the graphene can substantially affect the performance of the transistor realised in this way [5]. It was also found that *R*_C_ significantly affects the high-frequency characteristics of the transistors [7]. A number of works address this issue, such as patterning the graphene at the contact region with holes [8], mild oxygen plasma treatment [9], selecting the contact metal according to its work function [10], or post-process annealing [11]. However, these studies were performed with metal electrodes deposited onto the graphene layer. Sensitive layers based on carbon allotropes enable the realisation of simple and effective chemoresistive sensors. For such sensors, it is essential to maximise the sensitivity to the acting analyte; that is, the degree of change in resistance (impedance) upon the change in the concentration of the analyte should be as high as possible. The sensitivity of sensors can be affected not only by the resistance of the sensitive layer, but also by the contact resistance between the sensitive layer and the metal electrode structure. In the case of a high *R*_C_, the sensor’s sensitivity is lost. The arrangement of these sensors is usually reversed compared with that of the GFET. The electrode structures are first created using photolithography, and then the active layer of graphene is applied on top. Graphene-sensitive layers are used to realise sensors intended for a wide range of applications, such as sensors for monitoring environmental parameters and biosensors [12,13,14,15]. Compared with carbon nanotubes, graphene has a planar structure and, thus, graphene has a larger area of contact with the electrode structure. Graphene also creates a large area of contact with the electrode structure compared with carbon nanotubes [16], where the contact resistance will be influenced precisely by the amount and arrangement of carbon nanotubes. Therefore, it is more advantageous to use graphene to determine the parameters affecting the contact resistance. The contact resistance can thus be influenced by the materials used, the surface treatment of the electrode structure before the deposition of the sensitive layer, and the methods of deposition of the sensitive layer onto the electrode structure. The environment in which the electronic elements are located also influences the contact resistance, through both humidity [17] and temperature [11]. The highest quality graphene is prepared by either mechanical exfoliation [18] or epitaxial growth on SiC [19], but these methods are not suitable for high-throughput applications. For such a purpose, layers prepared by liquid (shear) exfoliation are characterised by very large yields but lower quality [20], which also result from cost-effective and environmentally friendly precursors [21]. As an in-between, the chemical vapour deposition (CVD) growth offers a compromise between coverage and performance [22]. The CVD process is, nowadays, accepted as the most suitable method used for the realisation of graphene-based large-area electronic devices; however, the sheet resistance and other electronic characteristics, such as carrier mobility, suffer from the disorder induced by both the growth and the subsequent polymer-assisted transfer onto the target substrate [23].

The transfer length method (TLM) in various modifications is the most frequently used technique to evaluate the contact resistance between graphene and the electrode structure [8,9,24]. This method makes it relatively easy to objectively and reliably determine the value of the contact resistance without the need for a complex evaluation of the influence of sheet resistance [10]. Gold/nickel, gold/platinum, and palladium are the materials commonly used for fabricating electrode systems. They are chemically stable and enable reaching *R*_C_ values below 100 Ωµm [24]. However, the cost makes these systems less favourable than, for example, copper, which is the most widely used metal in electrical engineering.

The aim of this work was thus to verify whether it was possible to achieve reasonably similar values of contact resistance for copper as for gold electrodes when pre-patterned before the deposition of single-layer (1L) CVD graphene. We also investigated how various surface treatment protocols of such an oxidising electrode system would affect the contact resistance and, finally, whether the contact resistance between gold and graphene could be further improved by its structural modifications, as suggested previously for better-defined combinations of higher quality graphene and metal electrodes deposited on top [8,9,10].

## 2. Materials and Methods

### 2.1. Materials

An electrode structure was designed for the experimental part, where the gaps between individual electrodes were set to 25, 40, 60, 80, 100, 200, and 300 µm (Figure 1).

The metal electrode structure based on a Cu or Au/NiCr layer was realised using photolithography on silicon carrier substrates (Si wafers) with a SiO_2_ layer. The sputtering deposition machine Balzers BAS 450 PM (Wordentec Ltd., Shebbear, UK) was used to deposit the Cu or Au/NiCr conductive layer. The thickness of the NiCr layer was 80 nm and the Au or Cu layer thickness was 150 nm. The exposition device Karl Suss MA 56 (Suss MicroTec SE, Garching, Germany) was used for the photolithography process, where the deposition of the resist ma-P1225 by Microresist Technology (Berlin, Germany) was carried out by spin coating. The thickness of the coated resist was 2 µm. As a carrier substrate, the 3-inch boron-doped silicon wafer from Nanografi (Ankara, Turkey) was used, with a thickness of 380 µm, a resistivity of 1–10 Ωcm, and a SiO_2_ thickness of 300 nm. At the end of the electrode structure realisation, the wet etching process was used for 30 s in the solution of HNO_3_, Ce(SO_4_)_2_, and H_2_O. The actual gaps between individual electrodes were measured using an Olympus confocal scanning microscope, LEXT OLS5000 (Olympus, Tokio, Japan). The measurement of the gap between the electrodes was carried out at three different places for a given gap, and the arithmetic mean was used for the precise evaluation of *R*_C_. The standard deviation of the measurement was ±0.96 µm for a gap of 25 µm and ±1.0 µm for a gap of 300 µm, which shows a negligible influence of the gap dimension heterogeneity on the evaluation of the resistance measurement results.

Before the actual deposition of the graphene, various methods of Cu electrode surface treatment were applied (Table 1). This involved mechanical cleaning, chemical cleaning in HNO_3_ (65% A.G., PENTA s.r.o., Prague, Czechia), or electrochemical polishing using H_3_PO_4_ (85% G.R., Lach-Ner s.r.o., Neratovice, Czechia). The mechanical cleaning (MC) of the electrodes consisted of fine grinding and polishing with the use of a rubber made from synthetic polymers (bis(2-ethylhexyl)hexahydrophthalate), during which oxides were removed from the surface of the electrodes. Chemical cleaning (CC) was performed using nitric acid in three different concentrations (4 vol.%, 18.5 vol.%, and 65 vol.%). Electrochemical polishing (ECP) of the electrode structure was carried out in a phosphoric acid solution (50 wt.%) upon applying 1 V vs. Ag/AgCl for 5 or 8 s [25]. The electrode system with Au was cleaned in the following sequence: 1. deionised (DI) water, 2. acetone (MicroChemicals GmbH, Ulm, Germany), and 3. isopropanol (MicroChemicals GmbH) for 10 min.

Graphene was grown by CVD in accordance with a previously established protocol [26,27,28]. Briefly, a 25 µm thick Cu foil with an area of the order of a few centimetres squared as a substrate was heated up to 1000 °C in an H_2_ atmosphere and annealed for 40 min. Afterward, the foil was exposed to a mixture of CH_4_ (1 sccm) and H_2_ (50 sccm) for 30 min. Additional annealing in H_2_ for 5 min was performed after exposure to the methane/hydrogen mixture. Total pressure during the entire CVD process was maintained at 350 mTorr. Finally, the substrate was cooled down under H_2_. The 1L graphene was transferred onto electrode structures using the nitrocellulose-assisted method [29]. Lists of all tested samples are provided in Table 1 (Cu electrodes) and Table 2 (Au electrodes). The defects in graphene after the transfer on Au TLM electrodes were created by oxygen plasma (PICO, Diener Plasma-Surface Technology, Ebhausen, Germany) with the following conditions. The chamber was pumped down to a base pressure of 0.2 mbar. Then, the pressure was gradually increased to 0.7 mbar with 99.999% O_2_ (Messer Technogas s.r.o., Prague, Czechia) using an in-built flow controller, after which the pressure was stabilised and the plasma was turned on at 35 W or 52.5 W RF generator for a specific treatment time of 10, 12, or 20 s (see Table 2). They were also created by Ar^+^ ion bombardment at 100 keV energy for different durations to reach irradiation doses in the range of 10^12^–10^13^ cm^−2^ (see Table 2). The samples were irradiated at the University of Helsinki with single-charged Ar^+^ ions using a 500 kV KIIA ion implanter from High Voltage Engineering Europa B.V. (Amersfoort, The Netherlands). Ions extracted from an ion source with a 20 kV voltage were analysed with a 90° magnet and accelerated with an 80 kV voltage. The samples were kept at room temperature during irradiation and the beam current was 50 nAcm^−2^. The beam current and total fluence were measured using a shoot-through Faraday cup with an aperture of 1.4 cm in diameter and a Faraday cup area of 1.5 cm^2^. For electron suppression, the Faraday cup had suppression electrodes at −500 V in front of and behind the Faraday cup. The defects in Ar^+^-bombarded samples were created only within the graphene in contact with the Au electrodes using direct-write photolithography patterning (MicroWriter ML3 Pro, Durham MagnetoOptics Ltd., Durham, UK) with an AZ^®^ ECI 3007 chemical resist (MicroChemicals GmbH) which, after illumination, was developed with an AZ^®^ Developer (MicroChemicals GmbH).

### 2.2. Methods

The TLM was used to determine the contact resistance values between graphene and metal electrodes. This technique consists of creating metal electrodes with different gaps on which the graphene layer is deposited. The resistance value is then measured between the individual contacts. The measured total resistance value (*R*_T_) is a combination of the contact resistance of graphene with the two neighbouring contacts (*R*_C_), the resistance of the graphene layer (*R*_G_) between the contacts, and the intrinsic resistance of the metallic layer (*R*_m_):*R*_T_ = 2*R*_m_ + 2*R*_C_ + *R*_G_(1)

Because *R*_C_ >> *R*_m_, the metal resistance is neglected. From the measured values, a graph of the total resistance values is constructed depending on the electrode distance. After the obtained linear dependence is extrapolated, the intersection with the *y*-axis expresses twice the value of the contact resistance (2*R*_C_). From the slope of the linear fit, the sheet resistance (*R*_S_) of graphene can be obtained, because *R*_G_ = *R*_S_(*L*/*W*), where *L* is the distance between the electrodes and *W* is the contact width. However, in this work, *R*_S_ was determined from the particular slope of measurement between contacts spaced at 100 and 200 µm, where the total resistance is dominated by the *R*_G_ and less so by the *R*_C_.

After the graphene was deposited on the electrodes, the resistance between the individual electrodes was gradually measured. The measurement of electrical resistance was carried out at room temperature and pressure, under standard illumination, using the four-point test method via the multimeter system Keithley DAQ6510 (Keithley Instruments, Solon, OH, USA) to eliminate the possible influence of supply wire resistance. The measurement accuracy of the resistance for the Keithley equipment was 0.01% of the reading and 0.001% of the range. This high measurement accuracy cannot cause a significant deviation in the measured total resistance values and, therefore, neither in the resulting *R*_C_ values. The measurement of contact resistance was carried out in DC mode concerning the TLM evaluation. Contact with individual samples was made using a homemade contact needle array prepared for the used electrode structure, for which the spring probes S1 Series from Smiths Interconnect (London, UK) were used. Samples 14 to 16 were measured in darkness using a Keithley 4200A-SCS parameter analyser (Keithley Instruments, Solon, OH, USA) in a two-point arrangement with gold probes (7-µm tip radius, Microworld, Grenoble, France), and *R*_T_ values were extracted from I-V characteristics. The electrical resistance was gradually measured between the individual electrodes (A-B, B-C, C-D, D-E, E-F, F-G, and G-H; see Figure 1). The first measurement was performed in the shortest possible time after the deposition of the graphene layer onto the electrode structure under normal ambient conditions (temperature 23 °C, relative humidity 30–60%). Subsequently, the samples were left in air under ambient conditions in the dark for 14 days, and then the electrical resistance between the individual electrodes was measured again, with the exception of where delays between sample preparation and measurement were longer that the 14-day gap (e.g., the Ar^+^ ion-irradiated samples). The effect of light on the contact resistance values for samples that were measured under laboratory lighting but kept in the dark, and for samples measured directly in the dark, was not observed. The measured values of the electrical resistance were plotted in a graph as a function of the gap between the individual electrodes.

Raman spectra of the transferred graphene (both pristine and with post-transfer defects) were measured using either a 532 nm laser excitation wavelength (with Alpha300R spectrometer, Witec, Ulm, Germany) or a 633 nm laser excitation wavelength (with a LabRAM HR spectrometer, Horiba Scientific, Lille, France) under a 100× objective (0.85 N.A.). The laser power was 1 mW under the objective. A grating of 600 lines/mm was used, giving a pixel-to-pixel resolution of ~1.8 cm^−1^. The spectrometer was calibrated using the F_1g_ mode of Si at 520.2 cm^−1^. The defect density, *n*_D_, and the average distance between defects, *L*_D_, in the 1L graphene were calculated using Equations (2) and (3) by using the intensity (amplitude) ratio of the D and G Raman bands (*I*_D_/*I*_G_) and the laser excitation wavelength (*λ_L_*) [30]:(2)$$L_{D}^{2}\;\left({\mathrm{nm}}^{2}\right)=\left(1.8\pm 0.5\right)\;\times \;10^{-9}\lambda_{L}^{4}{\left(\frac{I_{D}}{I_{G}}\right)}^{1}$$
(3)$$n_{D}\left({\mathrm{cm}}^{2}\right)=\frac{\left(1.8\pm 0.5\right)\;\times \;10^{22}}{\lambda_{L}^{4}}\;\left(\frac{I_{D}}{I_{G}}\right)$$

## 3. Results and Discussion

The core result of all individual experiments is the dependence of the resistance between every two neighbouring electrodes on the distance between these electrodes. The measured values for each sample were subsequently extrapolated by a least-squares linear fit, where the intersection with the *y*-axis determines twice the *R*_C_ value between the electrode structure and the graphene layer.

### 3.1. Cu Electrode System

In a copper electrode system, copper oxidation has a significant effect on contact resistance. For this reason, it is necessary to remove the oxide layer before graphene deposition. Samples with an applied graphene layer without any surface treatment of the copper electrodes were not realised and measured because the oxidised surface of the copper electrodes will significantly increase the contact resistance, thereby preventing any meaningful electrical measurement. Different methods of cleaning the copper electrode structures were compared to evaluate the efficacy of such treatments on the contact resistance between the copper electrodes and the graphene layer that was transferred on top of them immediately after the cleaning procedure. Figure 2A–D shows the dependence of the resistance on the electrode distance for different surface treatment methods.

The first striking observation, regardless of the particular cleaning method, is the large spread of the measured resistance between the samples. This indicates large sample-to-sample variation caused by imperfect removal of the surface copper oxides. Although some of the variations could be caused by the graphene transfer itself, the case of Au electrodes evidences only a small contribution of the transfer (see next section). Figure 2E shows individual *R*_C_ values for different methods of the surface treatment of copper electrodes. The lowest *R*_C_ values were achieved for the MC and CC, reaching hundreds of ohms on average. For the ECP, the contact resistance was the highest, in the order of kΩ units. These higher *R*_C_ values are probably due to the residual oxide layer that was not removed during this surface treatment. As can be seen in Table 1 and Figure 2B, the lowest *R*_C_ value was achieved for the longest electropolishing time—although it was still higher than for most of the other treatments. Hence, in contrast to the other treatment types, it would be necessary to set the conditions of the cleaning process according to the thickness of the oxide, making this method generally less viable for standardised utilisation. Concerning the other methods, the *R*_C_ values of hundreds of ohms indicate that, despite the removal of most of the copper oxide, no good electrical contact with graphene was realised. Such high *R*_C_ values are probably caused by a thin and heterogeneous layer of copper oxide forming instantaneously on the cleaned electrodes during the graphene transfer procedure. Furthermore, the oxide layer can grow even under the graphene layer due to the defects present in its structure, and the surface of the electrodes is thus still partially accessible to air. The air can also penetrate under the graphene layer from its edges. As shown previously, the graphene layer on the copper electrodes worsens the oxidation [31], thereby increasing the contact resistance. This effect is also proven by the increased *R*_C_ values during repeated measurements after 14 days of aging when the samples were left in an ambient environment (see the M1 and M2 columns in Table 1). The increase in *R*_C_ due to sample aging is in the order of tens of percent, reaching up to 200%, with an average 46% increase. However, the *R*_C_ increase phenomenon was not observed in samples on gold electrodes, where identical contact resistance values were measured even after aging of the samples, as described in the following sections.

### 3.2. Au Electrode System

For the Au electrode system, a negligible sample-to-sample variation was observed, in contrast to the Cu electrodes, despite only implementing a standard cleaning procedure with DI, acetone, and isopropanol before the graphene transfer. Figure 3A shows the results of the resistance measurement for graphene transferred onto the Au electrodes for two groups of samples, one with the transferring polymer (nitrocellulose) removed and one with the polymer left on top of the graphene. No substantial difference was observed in the *R*_C_ values between these two groups (Table 2 and Figure 3D). In the case of graphene layers with the polymer on top, the *R*_C_ values range from 5.7 Ω to 34.7 Ω. For samples where the polymer has been removed from the graphene, the *R*_C_ varies from 9.3 Ω to 26.1 Ω. The small spread of *R*_C_ values within the sample groups can be ascribed to the minor variations in the graphene quality over the 0.5 × 0.5 cm^2^ area and an uneven distribution of the omnipresent cracks, voids, and wrinkles induced by the transfer process [32,33]. The negligible effect of the polymer presence on *R*_C_ allows for a straightforward interpretation of the results in the following experiments, where parts of the graphene between the contacts were masked by a resist to prevent irradiation damage by Ar^+^ ions.

Figure 3B shows the dependence of resistance on electrode distances for samples where the graphene surface was treated with O_2_ plasma for various durations and powers to create defects in the graphene structure. The resulting *R*_C_ values are in the range of 28.7 Ω to 37.2 Ω (Table 2 and Figure 3D). In this case, the parts of graphene between the electrodes were not masked by the resist. Figure 3C shows resistance values as a function of electrode distance for graphene layers where defects were created by bombarding the layer with Ar^+^ ions of 100 keV energy. The *R*_C_ values for the samples treated in this way vary from 28.9 Ω to 35.1 Ω (Table 2 and Figure 3D). It can be seen from these values that the defects created in the graphene layers do not significantly affect the value of the contact resistance compared with the graphene layer without defects.

Recent studies have reported that defects created in graphene layers decrease contact resistance [8,9,34,35,36]. The effect is usually attributed to stronger covalent bonding of graphene and metal or a reduction in the bonding distance [37]. However, this phenomenon was not demonstrated in the tested samples. When comparing our experimental system with the above-mentioned studies, which were usually conducted on higher-quality graphene with the metal contacts deposited on top, we can identify two main reasons for the observed behaviour: (i) the deposition of graphene onto the already fabricated electrodes, which does not lead to intimate contact between the two counterparts, and/or (ii) the presence of growth- or transfer-induced defects in graphene before any extra treatment. Hence, further introduction of defects might not significantly modify the contact quality.

Table 2 shows the measured values of the contact resistances *R*_C_ for samples with varying treatments. The sheet resistance (*R*_S_) of the graphene layer was also determined for the samples. As expected [38,39], a minor increase in *R*_S_ can be observed for the plasma-treated samples. In conjunction with the Raman spectroscopy measurements, shown in Figure 4, this proves that the defects in graphene were indeed caused by the treatment. The intensity of the Raman D band at ~1350 cm^−1^ increases with the increasing treatment times and powers for the O_2_ plasma and with the total dose for Ar^+^ ion bombardment. At the same time, the intensity of the 2D band (at ~2700 cm^−1^) decreases, also corresponding with an increasing lattice disorder [30]. The measured intensity ratios of the D and G bands (*I*_D_/*I*_G_), and the ranges of defect densities (*n*_D_) and distances (*L*_D_) calculated using Equations (2) and (3), are summarised in Table 2. The *I*_D_/*I*_G_ varies from 0.11 to 3.94, with the corresponding *n*_D_ from the lower limit to the upper limit. We note that due to different laser excitation wavelengths used for the Raman measurement, only the ranges of *n*_D_ and *L*_D_ values should be taken as quantitative, not the average value, due to the approximation in Equations (2) and (3) [30]. Additional information on the nature of the defects can be obtained by analysing the other defect-related Raman band, D’, which appears as a higher-frequency shoulder of the G band at ~1620 cm^−1^. The intensity ratio of the D and D’ bands (*I*_D_/*I*_D’_) is related to the defect type [40,41]. In our case, for the samples with the highest defect concentration (i.e., number 13 for the O_2_ plasma and 14 for the Ar^+^ bombardment), the *I*_D_/*I*_D’_ ratios are ~20.6 and 5.5, respectively. These ratios indicate that most of the defects are sp^3^-like for the plasma-treated samples and vacancy- or boundary-like for Ar^+^ irradiated samples. Hence, regardless of the majority defect type, the contact resistance is still dominated by the graphene deposition on top of the electrodes. A visual comparison of the effect of the defect density is shown in Figure 5, where *R*_C_ is plotted against *n*_D_. The squares with colours corresponding to the spectra in Figure 4 represent samples with induced defects. They mostly fall within the bounds of *R*_C_ values for the non-modified samples, confirming that in our graphene–electrode system, defects with areal density up to ~6 × 10^11^ cm^−2^ do not influence contact resistance more than the differences caused by imperfect control of the graphene transfer process. We also note that regardless of the defect presence, the *R*_C_ values are stable over time, in contrast to the case of Cu electrodes (cf. Table 1 and Table 2).

As can be seen from the results presented in the above sections, the contact resistance between CVD graphene and a pre-patterned electrode beneath it is highly dependent on the particular contact metal. The *R*_C_ reached an average value of 1.25 ± 1.54 kΩ for Cu and only 0.03 ± 0.01 kΩ for Au, with the same electrode dimensions. In previous studies, where the metal electrodes were fabricated on top of the graphene samples, several reasons for the differences in *R*_C_ between graphene and various metals were suggested. The work function difference between graphene and the metal can cause differences in the charge doping of graphene, thereby changing the density of its states and, consequently, sheet resistance [42]. The work function of Au is ~5.1 eV and only ~4.9 eV for Cu; hence, with respect to the work function of graphene (~4.6 eV), the difference is smaller by 0.2 eV for the Cu–graphene pair [43]. Therefore, the doping of graphene is also smaller in the latter case, and the sheet resistance is larger. However, it also must be noted that the work function is strongly crystallographic face-dependent, and it can reach ~4.5 eV for Cu (100) or (110) faces [43]. In contrast, Watanabe et al. [44] reported no dependence of the metals’ work function on the *R*_C_ with graphene. Instead, they suggested that chemical cleaning and the microstructure of the deposited metal are the key factors influencing contact quality. In our case, however, the large difference between *R*_C_ values for Cu and Au, the heterogeneity between individual treatment types and samples on Cu electrodes, and the obvious worsening of *R*_C_ over time for Cu, all point to the major influence of the copper oxide removal and subsequent continuous growth. The Cu electrodes, in spite of the lower cost, are thus not favourable for the system in which they are pre-patterned under graphene.

## 4. Conclusions

We investigated the quality of electrical contacts between 1L CVD graphene and Au and Cu electrodes pre-patterned on the substrate before the transfer of graphene on top. It is clear from the obtained results that in the case of the graphene layer on the Cu electrode system, the type of cleaning of the electrode structure before the graphene transfer strongly affects the resulting contact resistance. Nevertheless, the lowest achieved contact resistance values were still in the order of hundreds of ohms. Such high contact resistance values can dampen any desired changes in the resistance of the graphene layer when it is utilised as the active layer in chemoresistive sensing applications. In the case of Au electrode systems, contact resistance in the order of tens of ohms was achieved without the need to clean the electrode system surface before the transfer. However, the contact resistance was not further improved in the Au–graphene system by modification of the graphene structure by defects with densities reaching 6 × 10^11^ cm^−2^, regardless of their type and origin. The achieved results can thus be the basis for the design of sensor elements based on graphene structures, as it is clear from these results which methods of modifying the electrode system and methods of graphene transfer have the greatest effect on the contact resistance and, thus, the resulting sensitivity of the sensor. Fabrication of pre-patterned electrodes can also be potentially utilised in other applications, where the graphene’s top surface is sensitive to post-processing, such as when manipulating its electrochemical or optical properties by molecules or nanoparticles.

## Figures and Tables

**Figure 1 nanomaterials-12-04444-f001:**
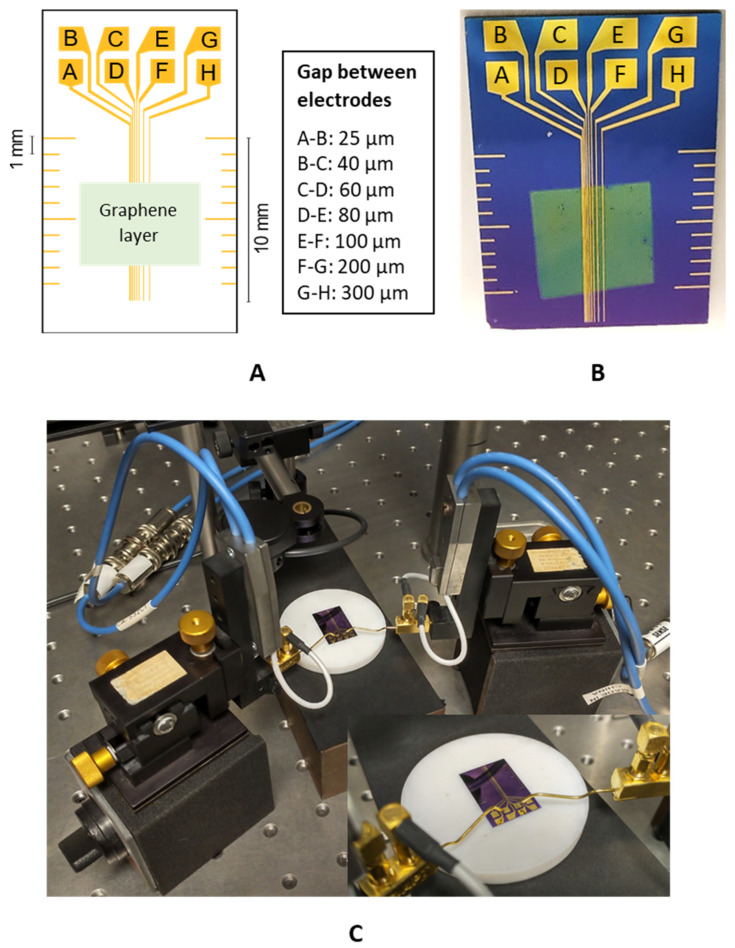
(**A**) Design of the electrode structure for the TLM; (**B**) realisation of the electrode structure on the Si wafer; (**C**) measurement configuration.

**Figure 2 nanomaterials-12-04444-f002:**
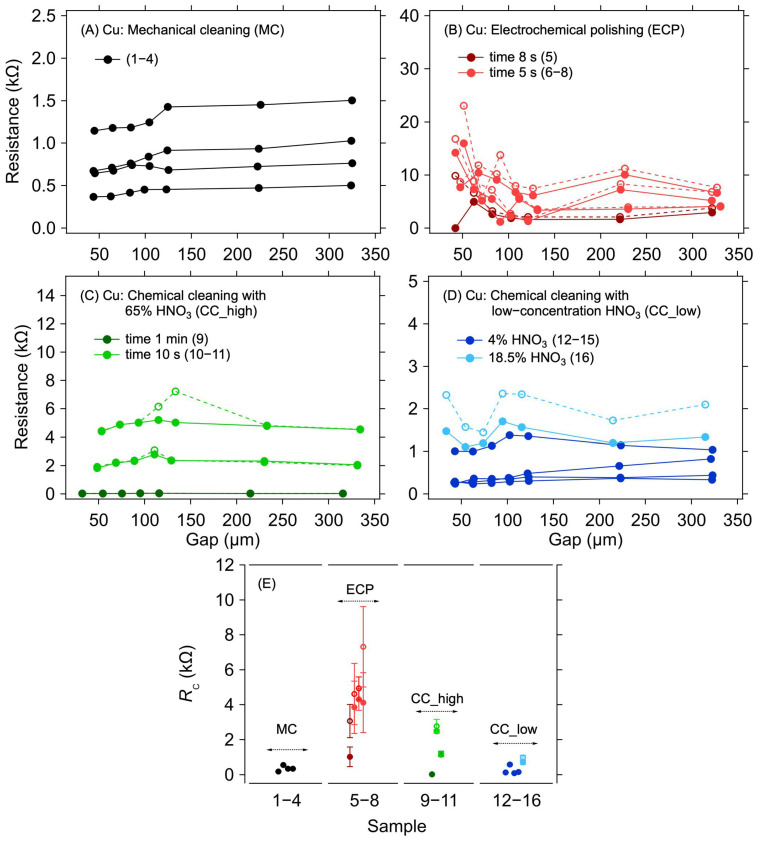
A 1L graphene–Cu electrode contact resistance for different Cu surface treatment methods. Resistance as a function of electrode gap for (**A**) mechanical cleaning, (**B**) electrochemical polishing: dark and light red colours for a polishing time of 8 and 5 s, respectively, (**C**) chemical cleaning in 65% HNO_3_: dark and light green colours for a cleaning time of 1 min and 10 s, respectively, and (**D**) chemical cleaning in low-concentration HNO_3_: dark and light blue colours for 4% and 18.5% concentration, respectively. The lines are guides for the eye only. (**E**) Contact resistance determined from the measurements shown in panels (**A**–**D**) with corresponding colours. The error bars in (**E**) reflect one standard deviation of the linear fits for the *R*_C_ determination. Solid and dashed lines and full and empty circles depict the first and second measurements (labelled as M1 and M2 in Table 1), respectively.

**Figure 3 nanomaterials-12-04444-f003:**
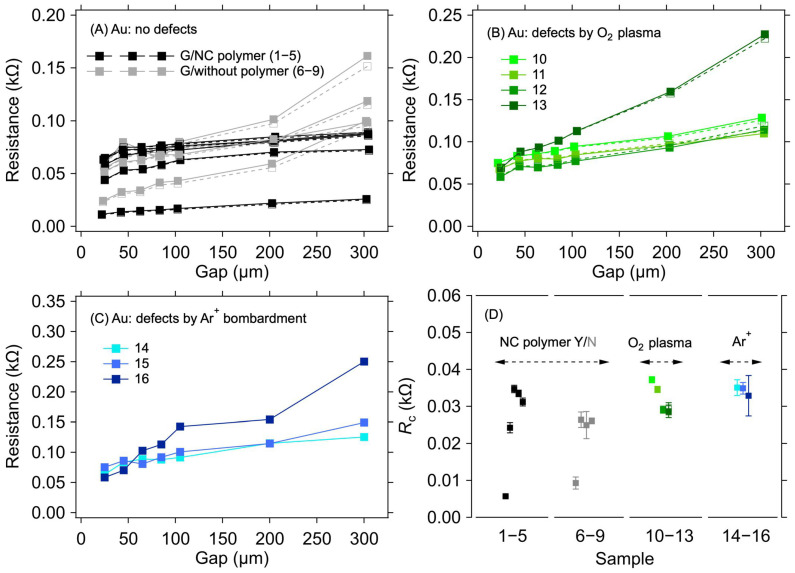
A 1L graphene–Au electrode contact resistance for different graphene manipulation methods. Resistance as a function of electrode gap for (**A**) as-transferred graphene with the transferring nitrocellulose washed away (black) or left on the graphene (grey), (**B**) graphene with defects created by the O_2_ plasma treatment, and (**C**) graphene with defects created by Ar^+^ ion bombardment. The lines are guides for the eye only. (**D**) Contact resistance is determined from the measurements shown in panels (**A**–**C**) with corresponding colours. The error bars in (**D**) reflect one standard deviation of the linear fits for the *R*_C_ determination. Solid and dashed lines and full and empty squares depict the first and second measurements (labelled as M1 and M2 in Table 2), respectively.

**Figure 4 nanomaterials-12-04444-f004:**
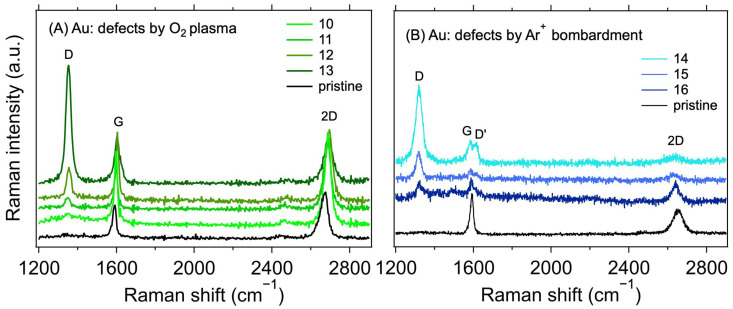
Raman spectra of 1L graphene transferred on Au electrodes with defects created using (**A**) O_2_ plasma (*λ_L_* = 532 nm) and (**B**) Ar^+^ ions (*λ_L_* = 633 nm). The black line corresponds to the as-transferred pristine graphene without additionally induced defects.

**Figure 5 nanomaterials-12-04444-f005:**
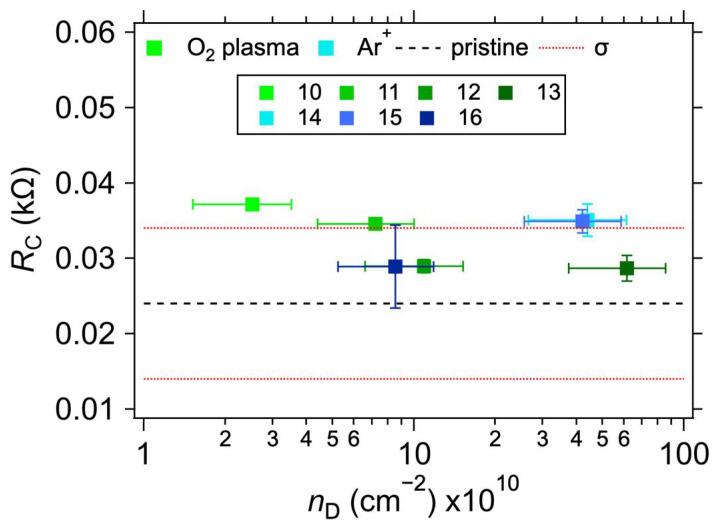
Contact resistance values for 1L graphene on Au electrodes as a function of defect concentration (*n*_D_). The black dashed and beige solid horizontal lines represent the average and one standard deviation (*σ*) of *R*_C_ values for the as-transferred 1L graphene on Au electrodes without added defects. The error bars reflect the uncertainties in *n*_D_ calculation from Equation (3) and one standard deviation of the linear fits for the *R*_C_ determination.

**Table 1 nanomaterials-12-04444-t001:** Tested samples of 1L graphene on the Cu electrode system. M1 and M2 are the first, immediate measurement, and measurement after 14 days in an ambient environment, respectively.

Sample		Electrode Surface Treatment	*R*_C_ (kΩ)	
			M1	M2
1	MC	rubbed with an eraser and cleaned in DI water	0.188 ± 0.009	-
2			0.561 ± 0.025	-
3			0.338 ± 0.019	-
4			0.335 ± 0.011	-
5	ECP	50% H_3_PO_4_/8 s/1V	1.024 ± 0.56	3.073 ± 0.955
6		50% H_3_PO_4_/5 s/1V	3.861 ± 1.500	4.618 ± 1.755
7			4.319 ± 0.635	4.945 ± 0.655
8			4.131 ± 1.705	7.320 ± 2.300
9	CC_high	65% HNO_3_/1 min, wash in DI water for 10 s	0.022 ± 0.002	0.022 ± 0.002
10		65% HNO_3_/10 s, wash in DI water for 10 s	2.475 ± 0.108	2.781 ± 0.389
11			1.142 ± 0.111	1.214 ± 0.139
12	CC_low	4% HNO_3_/10 s, wash in DI water for 10 s	0.127 ± 0.011	-
13			0.584 ± 0.059	-
14			0.087 ± 0.011	-
15			0.150 ± 0.015	-
16		18.5% HNO_3_/1 min, wash in DI water for 10 s, 1M HCl/1 min, and wash in DI water for 10 s	0.670 ± 0.108	0.981 ± 0.139

**Table 2 nanomaterials-12-04444-t002:** Tested samples of 1L graphene on the Au electrode system. The conditions for the O_2_ plasma etching denote treatment time/RF generator output power of 35 W (labelled as L) or 52.5 W (H). The note at the Ar^+^ bombardment corresponds to the total ion dose delivered to the sample. M1 and M2 are the first, immediate measurement, and measurement after 14 days in an ambient environment, respectively. The *I*_D_/*I*_G_ determination originates from Raman measurements using 532 nm and 633 nm laser excitation wavelengths for samples 10–13 and 14–16, respectively.

Sample	Polymer on Top	Defects		*I*_D_/*I*_G_	*L*_D_ (nm)	*n*_D_ (×10^10^ cm^−2^)	*R*_C_ (Ω)		Rs (kΩ/sq)
							M1	M2	
1	Yes	No			-	-	5.7 ± 0.2	5.7 ± 0.2	0.920
2							24.3 ± 1.4	24.1 ± 1.3	2.588
3							34.7 ± 1.0	34.6 ± 1.0	3.031
4							33.5 ± 0.9	33.4 ± 0.9	3.074
5							31.2 ± 1.1	31.2 ± 1.0	3.052
6	No						9.3 ± 1.6	9.2 ± 1.5	2.189
7							26.4 ± 2.1	26.3 ± 2.0	2.289
8							25.0 ± 3.7	25.2 ± 3.3	2.937
9							26.1 ± 0.7	26.0 ± 0.7	2.344
10	No	O_2_ plasma	12 s/L	0.11	35.5 ± 7.1	2.5 ± 1.0	37.2 ± 0.7	37.3 ± 0.7	2.290
11			20 s/L	0.32	20.9 ± 4.2	7.2 ± 2.8	34.6 ± 0.8	34.6 ± 0.8	2.173
12			10 s/H	0.48	17.1 ± 3.4	10.9 ± 4.3	29.0 ± 0.9	29.1 ± 0.9	3.488
13			20 s/H	2.73	7.2 ± 1.4	61.5 ± 24.1	28.7 ± 1.7	29.4 ± 1.6	2.324
14		Ar^+^ bombardment	3.6 × 10^13^ cm^−2^	3.94	8.5 ± 1.7	44.2 ± 17.4	35.1 ± 2.1	-	2.573
15			1.2 × 10^13^ cm^−2^	3.76	8.7 ± 1.7	42.1 ± 16.5	34.9 ± 1.6	-	2.476
16			4 × 10^12^ cm^−2^	0.76	19.3 ± 3.9	8.5 ± 3.3	28.9 ± 5.5	-	2.362

## Data Availability

The data presented in this study are available upon request from the corresponding author. The data are not publicly available due to IP protection revision of related studies.

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
