# Peer review of "Electrical Contact Resistance of Large-Area Graphene on Pre-Patterned Cu and Au Electrodes"

_nanomaterials, 2022, doi:10.3390/nano12244444_

Round 1
Reviewer 1 Report
The manuscript ID nanomaterials-2104264 is an original article mainly devoted to describe particular factors with an influence in the contact resistance exhibited by graphene samples and Cu or Au electrodes. Please see below a list of comments to the authors:
1. Advantages and disadvantages of the interaction between graphene and Au vs graphene and Cu must be better described and if possible presented in a table to easily visualize the value of the analyzed results.
2. The experiments were carried out in darkness? The light can influence the electrical effects exhibited by graphene-based samples.
3. It would be interesting to comment about the influence of the imaginary part of the electrical impedance in the electrical observations and not only to explain the effects considering the electrical resistance.
4. Why the authors only report electrical contact resistance? Most of the potential applications of carbon/metal systems require the study of frequency dependent parameters. The authors are invited to describe perspectives and future implications. You can see for instance: https://doi.org/10.1364/OE.27.007330
5. A confrontation of the main findings with updated publications in the topic must be included to highlight the importance of the conclusions. You can see for instance: https://doi.org/10.1016/j.diamond.2012.01.019
6. The authors report the main experiment in 3 different areas of the sample. But proper reproducibility and statistics of the experimental section ought to be reported in order to see that the results are systematic instead of incidental.
7. Please include more details about the physical mechanisms responsible for the differences in the coupling between graphene and the metals studied to explain the differences in the data plotted in figures 2 and 3.
8. A photo of a representative measurement would be welcome.
9. It is recommended to split and present the citations in individual form instead of using the collective citation form. This is in order to better justify the panoramic state of the art of the topic studied.
10. The error bar in experimental data should be provided.
Reviewer 2 Report
Dear Authors
The manuscript is focused on determining factors which influence contact resistance between graphene prepared by chemical vapour deposition and pre-patterned Cu and Au electrodes, onto which graphene was subsequently transferred.
The following suggestion and comments should be taken:
1. The authors could insert more numerical data into the Abstract for enhancement of the manuscript.
2. The overall English needs to be improved. Please seek guidance from a native English speaker if possible ("the" "a", commas, plural form and others could be corrected).
3. The introduction section needs enhancement few sentences about graphene, modification with heteroatoms, and potential applications. Please cite: (1) Polymers 2020, 12(10), 2189; https://doi.org/10.3390/polym12102189 (2) Materials 2020, 13(21), 4975; https://doi.org/10.3390/ma13214975 (3) Graphene materials from microwave-derived carbon precursors 2021 https://doi.org/10.1016/j.fuproc.2021.106803
4. Figure 2. Please correct this image for better quality (the inscriptions).
5. Figure 3. Please correct this image for better quality (the inscriptions).
6. Could the authors include the standard deviation of the used methods?
7. Could authors add SEM or HRTEM images of obtained materials?
8. Please add to the results part in the Raman spectra - ID/IG ratio and comments to the text.
9. Why author choose graphene not other carbons for the study? Please explain in the comments.
10. Authors are suggested to describe some future applications in conclusions.
Round 2
Reviewer 1 Report
In my opinion, the results worth publication regarding they can be useful for future research in a relevant topic. Most of the points raised in the review stage have been addressed and clarified. Then I can recommend this work for publication in present form.
Reviewer 2 Report
Dear Authors
I recommend this manuscript for publication.